# Dance of The Golgi: Understanding Golgi Dynamics in Cancer Metastasis

**DOI:** 10.3390/cells11091484

**Published:** 2022-04-28

**Authors:** Rakhee Bajaj, Amanda N. Warner, Jared F. Fradette, Don L. Gibbons

**Affiliations:** 1Department of Thoracic/Head and Neck Medical Oncology, The University of Texas MD Anderson Cancer Center, 1515 Holcombe Blvd, Houston, TX 77030, USA; rbajaj@mdanderson.org (R.B.); awarner2@mdanderson.org (A.N.W.); jjfradette@mdanderson.org (J.F.F.); 2UTHealth Graduate School of Biomedical Sciences, The University of Texas MD Anderson Cancer Center, 1515 Holcombe Blvd, Houston, TX 77030, USA; 3Department of Molecular and Cellular Oncology, The University of Texas MD Anderson Cancer Center, 1515 Holcombe Blvd, Houston, TX 77030, USA

**Keywords:** Golgi, secretion, cancer secretome, metastasis, invasion, EMT, cancer therapeutics, biomarkers

## Abstract

The Golgi apparatus is at the center of protein processing and trafficking in normal cells. Under pathological conditions, such as in cancer, aberrant Golgi dynamics alter the tumor microenvironment and the immune landscape, which enhances the invasive and metastatic potential of cancer cells. Among these changes in the Golgi in cancer include altered Golgi orientation and morphology that contribute to atypical Golgi function in protein trafficking, post-translational modification, and exocytosis. Golgi-associated gene mutations are ubiquitous across most cancers and are responsible for modifying Golgi function to become pro-metastatic. The pharmacological targeting of the Golgi or its associated genes has been difficult in the clinic; thus, studying the Golgi and its role in cancer is critical to developing novel therapeutic agents that limit cancer progression and metastasis. In this review, we aim to discuss how disrupted Golgi function in cancer cells promotes invasion and metastasis.

## 1. Introduction

Protein trafficking and secretion are cellular processes that are vital for diverse functions, such as hormone release, defense, growth, cellular migration, and cell homeostasis [1]. Proper protein trafficking—from the endoplasmic reticulum (ER) to the plasma membrane (PM), various target organelles, or the extracellular (EC) space—is required for cell survival [2]. Alterations in protein secretion can lead to numerous diseases, including obesity, diabetes, chronic inflammation, and cancer [3]. Several mechanisms of secretion are involved in trafficking proteins, including conventional protein secretion (CPS) via the Golgi and the ER, and unconventional protein secretion (UPS) via organelles, such as autophagosomes or lysosomes. Here, we focus on Golgi-mediated secretion; readers are referred to other reviews of unconventional protein secretion [4,5,6].

Proteins that contain a specific transmembrane domain or a leader sequence are transported to the Golgi apparatus where the organelle acts to modify, sort, and traffic the proteins to their destined locations. The Golgi apparatus, originally coined the “apparato reticular de Golgi” [7] will henceforth be referred to as the Golgi in this review. Briefly, proteins synthesized in the ER travel in an anterograde direction through the *cis*-, *medial*-, and *trans-*Golgi compartments. Trafficked proteins may undergo post-translational modifications (PTM) in these compartments, where PTMs, such as glycosylation or proteolytic cleavage, act as signals for specific receptor interactions that direct the spatial fate of the protein [8,9,10,11]. Secreted proteins help to form the extracellular matrix (ECM), interact with or modify the ECM, or mediate signaling to other cells. The Golgi also regulates the retrograde transport of proteins from the PM to the Golgi, or from the Golgi to the ER. ER-resident chaperone proteins, such as calreticulin and heat shock proteins, contain a KDEL ligand that binds to the Golgi KDEL receptor (KDELR), facilitating the return of these proteins back to the ER. Retrograde trafficking allows for the recycling of proteins back to their original site of function [12]. For a more in-depth explanation of ER–Golgi protein trafficking, readers are referred to the following references [1,10,11].

Golgi dynamics are a finely-tuned process, with dysregulation leading to various pathologies, including cancer [3]. Golgi-associated genes are commonly mutated in cancer and were linked with enhanced metastasis and lower patient survival [13]. Metastasis accounts for a large percentage of cancer-related deaths, where metastatic cells disseminate from the primary tumor, circulate throughout the body, and colonize secondary organs [14]. For example, in non-small cell lung cancer (NSCLC), chromosome 1q21-43, which contains many Golgi-related genes involved in trafficking, is frequently amplified [13,15]. This suggests the importance of the organelle in cancer progression and survival. The functional alterations of the Golgi regulate the cancer cell secretome, the set of all proteins and lipids secreted into the EC space [16]. Different types of cells or certain pathological conditions result in a unique secretome. Cancer cells in particular have a distinct secretome, which is partly due to altered ER–Golgi trafficking and dynamics that aid in their invasive and metastatic biology. There are now various ways to visualize and understand the Golgi and its functions, including cryo-electron microscopy and tomography to study the structural details of the organelle [17]; live-cell confocal light microscopy to investigate the dynamic nature of the Golgi [18]; fluorescence recovery after photobleaching (FRAP) assay to determine the state and connectivity of Golgi ribbon linking and cisternal stacking [19]; and secreted reporters, such as the secreted alkaline phosphatase (SEAP) or Gaussia Luciferase (Gluc) to monitor the secretory pathway and quantify secretion [20]. Immunoelectron microscopy was used to detect mislocalization of a cis-Golgi marker to the rough endoplasmic reticulum in colon cancer cells and tissues compared to normal colon cells suggesting disruption of Golgi morphology in cancer [21]. Other electron microscopy images indicate that in vivo colorectal cancer cells have tight clusters of Golgi stacks, with fewer vesicles at the *trans*-Golgi, swollen Golgi cisternae, and no clear orientation of the Golgi stacks compared to normal cells [22]. These tools to study the Golgi are used to enhance our understanding of how the Golgi and Golgi-mediated secretion alter the metastatic cascade. This holds potential for improving patient prognosis through the development of diagnostic and therapeutic tools. 

## 2. The Golgi Alters the Cancer Cell Secretome, the Tumor ECM, and Immune Surveillance

The Golgi is responsible for the secretion of components that help form the ECM; hence, alterations in Golgi dynamics of cancer cells can make the secretome and the ECM more malignant (Figure 1). The ECM composition of proteins, proteoglycans, glycoproteins, and water varies between tissue types or pathological conditions, such as cancer [23]. Connective tissue and the basement membrane (BM) are also part of the ECM [24]; they provide structural support, survival signals, and modulate crosstalk between cell types [25]. Cellular signaling pathways regulating adhesion, migration, or survival can be affected by direct cell–ECM interactions or via cytokines and growth factors that partition between ECM binding and the interstitial fluid. ECM remodeling occurs through mechanisms that maintain proper ECM composition, structure, and degradation. For example, matrix metalloproteases (MMPs) can break down the ECM components, while proteins, such as lysyl oxidases, can alter ECM cross-linking [23]. Under normal cellular conditions, the Golgi secretes components such as cytokines, growth factors, and proteoglycans that help form the ECM [26]. The ECM collagen component is modified in the Golgi as smaller, immature procollagen units before secretion into the EC space. After secretion, the pro-collagen units are assembled into collagen to form the ECM. The secretion of other ECM components, such as fibronectin and tenascin-C, were proposed to be secreted by the Golgi; however, these studies remain inconclusive regarding their method of exocytosis [23]. 

Cell–cell and cell–ECM interactions help to prevent malignant cell transformation by maintaining appropriate homeostatic signaling. Increased secretion of ECM components, often termed the desmoplastic stroma, and enhanced ECM remodeling frequently occur in many tumor types [16,23]. Some of the molecules secreted into the EC space via the Golgi that can prompt these changes are MMP-2, MMP-9, and the membrane type 1 MMP (MT1-MMP) [27,28]. MMPs can modulate metastasis by breaking cell adhesion molecules and removing ECM barriers to promote invasion [29]. Additionally, MMPs can cleave and activate growth factors, such as VEGF, to promote cancer metastasis by enabling angiogenesis [14]. Golgi protein 73 is a transmembrane protein that, when secreted, can promote the transcriptional upregulation of MMP-13 through the cAMP-responsive element-binding (CREB) protein, which contributes to hepatocellular carcinoma and metastasis [30]. MMP-2 and MMP-9 are important for the degradation of collagen IV, which is present in the BM [24]. Breakdown of the BM is necessary during various steps of the metastatic cascade, including primary tumor cell dissemination, intravasation, extravasation, and colonization at the secondary site. Besides MMPs, many other proteins can be secreted by the Golgi that aid in metastasis. For example, a Golgi-associated protein, namely, phosphatidylinositol-4-kinase IIIβ (PI4KIIIβ), is necessary for the secretion of pro-survival and pro-metastatic factors—such as clusterin, semaphorin-C, lysyl hydroxylase-3, tissue inhibitor of metalloproteinase-1, and peroxiredoxin-5—suggesting that Golgi-related elements can drive secretion of proteins that further alter the tumor microenvironment to promote metastasis [13].

The Golgi can also enhance the secretion of immune factors that aid in the creation of an immunosuppressive tumor microenvironment (TME) to promote tumor progression and metastasis. PI4KIIIβ also mediates exocytosis of certain immune-modifying molecules from lung adenocarcinoma cells—such as CXCL1, IL-1α, IL-8, and VEGF—that aid in the migration of myeloid-derived suppressor cells, which is a key immunosuppressive cell type associated with metastasis [13,31]. Furthermore, the Golgi membrane protein, GOLPH3 interacts with exosome-localized cytoskeletal-associated protein 4 to enhance secretion of Wnt3a, which plays a role in limiting T cell differentiation [32,33]. This secretion helps promote stem-like phenotype and metastasis in NSCLC. Besides tumor cells, other non-tumor stromal cells also secrete factors that are incorporated into the cancer secretome. Inflammatory cells are a major subset of the TME that secrete cytokines and chemokines thereby promoting chronic inflammation, which is a hallmark of cancer [34,35]. For example, upregulation of a Golgi-associated protein, namely, golgin-245, in macrophages enhances exocytosis of the inflammatory cytokine, namely, tumor necrosis factor-α (TNFα) [36]. The NF-κB transcription factor family is activated in B cells by cell membrane receptor stimulus. Once activated, NF-κB then stimulates Golgi-mediated exocytosis of I-CAM1, TNF-α, IL-1, IL-8, IL-6, and cyclooxygenase-2 (COX-2), all of which are known drivers of tumor-proliferating inflammation and metastasis [37]. Specifically, IL-6 can promote pro-inflammatory signaling that is linked to cancer progression; it is highly expressed in many cancer types, including colorectal, breast, lung, and pancreatic cancer; correlates with poor survival, and aids in EMT-mediated cancer cell invasion [38,39]. Cancer cell and tumor-associated macrophage (TAM) interactions can promote the deposition of angiogenic and lymphangiogenic growth factors, cytokines, and proteases into the ECM. More specifically, TAMs enhance the secretion of growth factors, such as epidermal growth factor (EGF), 14-3-3 zeta protein, MMPs, VEGF, CCL2, and CXCL8 [34,35]. Thus, the Golgi has widespread functions in modulating not just the ECM but also other aspects of the TME, such as the immune landscape, to drive cancer metastasis.

## 3. Golgi Dynamics in Cancer Metastasis

### 3.1. Golgi-Mediated Vesicular Trafficking and Exocytosis

Vesicular trafficking was described as the “intracellular highway to carcinogenesis” [40]. Proper Golgi-mediated trafficking in cells maintains the steady flow of proteins from the Golgi to other organelles, the cell surface, and the EC space. Cancer cells hijack this process to exocytose proteins that eventually remodel cancer cells and their TME for metastasis [40]. RAB and ARF GTPases, which are frequently mutated in cancers, denote the vesicle and organelle identity and are required for the correct delivery of vesicles to their target organelle. Specifically, the activity of the small GTPases RAB and ARF are regulated by guanine exchange factors (GEFs) and GTPase activating proteins (GAPs) to maintain their GTP-bound active or GDP-bound inactive states, respectively. Both RAB and ARF proteins function through their effectors to regulate various functions, such as cargo budding, vesicle fusion, and exocytosis [41]. Aberrant activity of GTPases or their effectors impairs vesicular trafficking in tumor cells through incorrect localization of vesicles and is associated with altered trafficking kinetics, tumorigenesis, and metastasis [40,42,43].

RAB proteins localize to specific organelles or vesicles and their GTPase activity is required for proper vesicular transport and fusion, while improper activation or expression of RAB proteins can help to promote cancer progression [41]. Localization of RAB40b on VAMP4-positive secretory vesicles mediates the secretion of MMP-2 and MMP-9 from the Golgi to the EC space and was shown to enhance the invasive potential of human breast cancer cells [44]. Additionally, the expression of RAB27b is correlated with lymph node metastasis in ER+ breast cancer patients as it promotes the secretion of HSP90α, where HSP90α acts as a chaperone that prompts MMP-2 cleavage and activation [45]. This suggests that the secretion of chaperone proteins by cancer cells could also enhance metastasis through stabilization and activation of ECM-degrading proteins. Moreover, RAB-GTPases also interact with cytoskeletal proteins, such as the motor proteins myosin and kinesin, to coordinate the directed movement of the vesicles along actin filaments and microtubules, respectively, to the EC space [40]. Golgi-driven exocytosis is regulated by RAB11 vesicles interacting with MyosinVa and -Vb to promote vesicle transport [46,47]. RAB11 has oncogenic roles in a variety of cancer types through altering vesicular trafficking, receptor recycling, and signaling pathways that control proliferation and metastasis [48,49]. RAB6 regulates vesicle movement by interacting with the kinesin KIF20a [50]. Through its effector PKA, RAB13 prevents actin polymerization and tight junction integrity, thereby promoting prostate cancer [51,52]. However, RAB13 was also implicated in disrupting cancer cell growth as loss of the protein inhibited junction proteins, Claudin-1 and Occludin [51]. More detailed mechanistic studies are needed to determine how RAB proteins promote cancer progression to develop therapeutic strategies to inhibit the GTPases or their interacting partners.

Similar to RAB functions, ARFs, ARF-GEFs, and ARF-GAPs regulate retrograde and anterograde transport in cells; hence, alterations in these proteins during cancer lead to enhanced vesicular transport, exocytosis, cellular invasion, and metastasis [53,54]. For example, ARF1 regulates Golgi exocytosis by altering cytoskeletal proteins such as myosin and F-actin. ARF1 mediates RhoA and RhoC activity, which regulates myosin light-chain phosphorylation and detachment of membrane-derived vesicles [55]. It negatively correlates with breast cancer patient survival [56]. Additionally, ARF GTPases also regulate retrograde transport that eventually alters the cancer cell secretome via a feed-forward mechanism, whereby enhanced retrograde trafficking stimulates anterograde trafficking. For example, ARF4 functions with the COPBI protein involved in Golgi-to-ER retrograde trafficking to enhance anterograde trafficking and breast cancer metastasis in murine models [57]. Additionally, ARF4 and COPBI are necessary for the secretion of pro-metastatic factors, such as VEGF, CXCL1, CXCL10, and CCL20 [57,58]. One of the key players in modulating retrograde transport with GTPases is the KDELR family. When KDELR binds ER chaperone proteins that have escaped to the Golgi, it stimulates retrograde transport of these chaperones back to the ER. This interaction of KDELR at the Golgi membrane activates Src-mediated anterograde vesicular transport, although in an unclear way. Thus, KDELR enhances Src-mediated anterograde transport, exocytosis of MMPs, ECM degradation, and invasion into melanoma cells [59,60]. These studies prompt future research to examine the molecular mechanism behind how retrograde trafficking stimulates anterograde transport and its implications in cancer biology.

Vesicular trafficking requires membrane fusion events, such as movement through the Golgi or upon vesicle fusion with the PM. Membranes contain proteins, termed SNARES, that can either be on the vesicle (v-SNARE) or the target membrane (t-SNARE). Once the v- and t-SNARES have fused, ATP is hydrolyzed by N-ethylmaleimide sensitive factor (NSF), which interacts with soluble NSF attachment factors (SNAPs) to release the SNARE proteins from the newly synthesized membrane [61,62]. SNARE and SNAP proteins are altered in cancer cells to promote invasion. VAMP7, which is a v-SNARE residing in the ER–Golgi membranes, co-localizes with SNAP23 and Bet1, which is an ER–Golgi SNARE, to aid in MT1-MMP translocation to invadopodia in invasive breast cancer cell lines [28,63]. Various SNARE proteins are expressed at early or later cancer stages and were proposed for use as biomarkers [64]. Syntaxin-6 is a t-SNARE on the trans-Golgi membrane that is highly expressed in renal cell carcinoma [65]. Here, it interacts with microtubules to direct EGFR to the Golgi, which is required for its nuclear translocation and activation [66]. Syntaxin-6 also co-localizes with VAMP4 and ATP11B to secrete cisplatin from the Golgi to be released into the extracellular space in ovarian cancer cell lines [67]. In synovial sarcoma cells, SNAP23 and VAMP-3 aid in the secretion of IL-6 and TNF-α, which are known to have pro-tumorigenic roles [68]. Additionally, αSNAP negatively regulates AMPK, which is a known factor in tumor progression [69], and promotes disassociation of cellular junctions, which was suggested to promote invasion [70]. SNARE and SNAP proteins mediate many functions of Golgi vesicular transport to promote cancer invasion, progression, and drug resistance.

Beyond proteins involved in vesicular trafficking, Golgi-specific proteins can act as oncoproteins that have global implications on Golgi dynamics, function, and secretion. Golgi phosphoprotein 3 (GOLPH3) is involved in regulating directed vesicle exocytosis, cell migration, Golgi morphology and orientation, and protein glycosylation. Upon interacting with various Golgi membrane and structural proteins, GOLPH3 forms the PI(4)P/GOLPH3/MYO18A/F-actin complex that provides the force required for proper vesicle budding and Golgi-to-PM trafficking, whereas loss of the complex interrupts transport. As GOLPH3 binds to both a Golgi protein, namely, PI(4)P, and a myosin family member, namely, MYO18A, it allows for trafficking of the Golgi vesicles along actin filaments to aid secretion from the Golgi. Hence, increased GOLPH3 expression, as observed in various cancers, including melanoma and NSCLC, enhances anterograde trafficking from the Golgi and leads to increased exocytosis of pro-metastatic factors, such as cytokines, growth factors, and Wnt molecules [32,71]. Additionally, Rizzo et al. showed that GOLPH3 can also control the trafficking and recycling of enzymes that enhance glycosphingolipid synthesis and abundance [72]. A GOLPH3-mediated increase in these enzymes upregulates cellular growth and proliferation via mitogenic signals [72]. Thus, this data indicates that, in addition to mediating trafficking, Golgi proteins might also modulate cellular signaling. Furthermore, we demonstrated that two Golgi-associated proteins, namely, inositol monophosphatase domain containing 1 (IMPAD1) and KDELR2, also enhance Golgi-mediated exocytosis of MMPs to drive lung cancer metastasis. Both IMPAD1 and KDELR2 negatively correlate with NSCLC patient survival [60]. These studies imply that other proteins with functions related to increased vesicular trafficking in both the anterograde and retrograde directions can enhance Golgi exocytosis to drive a malignant TME during metastasis.

Furthermore, cellular conditions, such as hypoxia and nitric oxide (NO) abundance, can also regulate trafficking in malignant cells [73,74]. Hypoxia, which is the depletion of oxygen, is a key hallmark of metastasic cells [75]. Arsenault et al. showed that under hypoxic conditions the proprotein convertase, namely, Furin, was translocated from the trans-Golgi to the endosome and PM by RAB4 [76]. At the PM, Furin processed proproteins that were involved in tumorigenesis. Interrupting this re-localization suppressed hypoxia-induced invasion of fibrosarcoma cancer cells. Hypoxia was also shown to promote an adaptive unfolded protein response (UPR) and a suppression of coatomer protein complex genes (COPA, COPE, and COPG), thereby hampering ER-to-Golgi trafficking via the JNK pathway [77]. Additionally, oxygen levels can affect NO availability in the Golgi by modulating NO synthase expression. NO is a critical regulator with dual functions in metastasis. It was implicated in cytotoxic death of liver, breast, and skin tumors, but can also promote angiogenesis and intravasation in a context-dependent manner [78]. NO affects the nitrosylation of Golgi proteins, such as NSF, which is involved in fusion events; thus, delaying ER–Golgi vesicular trafficking [79]. Hence, cellular homeostasis is required to maintain normal trafficking and prevent tumorigenesis.

### 3.2. Golgi Orientation Governs Cell Polarity and Directional Migration

One of the hallmarks of metastatic cells is their ability to transition from an apico-basal polarity to a front–rear/migratory polarity, allowing them to spread from the primary tumor to secondary locations [14]. Similarly, the Golgi regulates its orientation to alter its function, where the orientation of the Golgi, with respect to the centrosome and microtubules, establishes the leading edge of a cell in the front–rear polarity model [71]. The secretome can be altered based on intracellular localization of the Golgi. For example, if the Golgi is closer to the cell membrane, such as in some cancer cells, it can enhance the frequency of secretion [71]. Additionally, Golgi orientation not only controls exocytosis but also guides vesicular flow to the leading edge of the cell to prompt directional migration [80]. The Golgi reorients to the leading edge of a cell, where migration of a cell begins, under stimuli, such as wounding or a chemotactic gradient in normal conditions [81]. It could be expected that these signals would stimulate Golgi reorientation in cancer cells as well; however, the stimuli and mechanisms for Golgi reorientation are diverse depending on the cancer type. For example, the small, secreted molecule gastrin, which aids in regulating gastric acid secretion, is abnormally expressed in pancreatic cancer. Gastrin stimulates phosphorylation of paxillin, which is a focal-adhesion-associated kinase protein, prompting Golgi reorientation to the leading edge to promote migration of pancreatic cancer cells [82]. Mutant LKB1, which is a tumor suppressor, disrupts Golgi reorientation to the leading edge through loss of interaction with CDC42, which is a Rho GTPase, affecting polarity in NSCLC cells [83]. Enhanced trafficking and cell motility at the leading edge of a cancer cell aids in the beginning of the metastasis process.

Several Golgi proteins are involved in altering cell polarity through their ability to regulate Golgi orientation. For example, Xing et al. showed that not only does GOLPH3 regulate exocytosis, as previously described, but also applies a tensile force through its interaction with PI(4)P/MYO18A/F-actin to promote a front–rear polarization by relocating the Golgi and trafficking toward the leading edge [84]. They also elucidated that this reorientation promotes directional cell migration, invasion, and metastasis through enhanced exocytosis. Thus, GOLPH3 is a key Golgi protein that has a multi-faceted role in regulating Golgi dynamics, further demonstrating that Golgi functions are complex and intertwined. Another Golgi matrix protein, namely, GM130, leads to a loss of polarity proteins, such as CDC42 and E-cadherin, when inhibited [85]. Additionally, they demonstrated that GM130 is necessary for the invasion of breast cancer cells [86,87]; thus, indicating an association between Golgi proteins, cell polarity, and cancerous behavior. Furthermore, paxillin also reorients the Golgi toward the leading edge to direct trafficking and cell migration through interactions with the centriole and is necessary for proper anterograde vesicular trafficking [88]. In melanoma, the interaction between paxillin, focal adhesion kinase (FAK), and histone deacetylase 6 enhanced the acetylated microtubule landscape, leading to the formation of invadopodia that promote cell migration, invasion, and metastasis [89]. The authors also showed that inhibiting this interaction may be a possible treatment for metastatic melanoma and it suggests that Golgi orientation is regulated by proteins with diverse functions [89]. Golgi positioning is dependent on cytoskeletal proteins, as Hela cells with knockdown of golgin proteins, namely, GMAP210 and golgin-160, that link the Golgi to motor proteins had altered directional secretion and migration due to random Golgi orientation despite intact microtubule and actin filaments [90]. Overall, Golgi localization within the cell, assisted by cytoskeletal factors, can alter the direction of Golgi-mediated secretion and promote cancer metastasis.

### 3.3. Morphology of the Golgi Apparatus

It is not only the orientation of the Golgi but also the structural morphology of the organelle that governs protein secretion during metastasis. The Golgi is a dynamic organelle that alters its structure to accommodate the physiological state of the cell. There are three main states of the Golgi—condensed, elongated, and fragmented—between which, the organelle oscillates in a context-dependent manner in different cancers (Figure 2).

The epithelial-to-mesenchymal transition (EMT) is one of the cellular phenomena that repositions and restructures the Golgi to redirect traffic toward the leading edge of cells, enabling migratory and invasive behavior [91]. EMT is hypothesized to allow cancer cells to gain the ability to disseminate from primary tumors and metastasize to secondary locations [92,93]. Our lab and others showed that metastasis-prone cells tend to be more plastic as they transition between epithelial and mesenchymal states [94]. Using NSCLC models, we also demonstrated that this transition is highly dependent on the negative-feedback loop between key regulators of a mesenchymal or epithelial state—namely, ZEB1 and miRNA-200 family (miR-200)—where they promote and inhibit EMT, respectively [94,95]. ZEB1-induced EMT leads to compaction and polarization of the Golgi with improved ribbon linking and cisternal stacking [91]. This organization is orchestrated by progestin and adipoQ receptor family member 11 (PAQR11), which is a scaffold for multiple Golgi-associated protein complexes that can then control organelle structure. The upregulation of PAQR11, which is correlated with shorter survival in cancer patients, directs and enhances protein exocytosis. In the same model, the authors show that PAQR11 also drives lung cancer metastasis, thus indicating a possible correlation between Zeb1-induced EMT, alterations in Golgi architecture, and the metastatic secretome [91]. Collagen deposition and crosslinking in the ECM are upregulated during EMT, which, in turn, enhances integrin/FAK/Src signaling in cancer cells to promote invasion and metastasis [93,96]. Additionally, enzymes such as lysyl hydroxylases that modify collagen to allow for its cross-linking and stabilization correlate with lung cancer metastasis and worse patient survival [97]. Moreover, increased collagen deposition also suppresses effective immune surveillance of the tumor immune microenvironment (TIME) by leading to an exhausted CD8+ T cell state [98]. Interestingly, the immune suppression was reversible upon inhibiting collagen interactions with the T cells, leading to an enhanced anti-tumor immune response. Hence, a feedback loop driven by EMT facilitates cellular secretion and ECM alterations that make cancer cells more malignant and resistant to therapy.

In contrast to the EMT-regulated Golgi compaction model that enhances exocytosis and malignancy, an elongated Golgi ribbon structure was also implicated in metastasis. GOLPH3 alters vesicular trafficking and Golgi orientation, as previously described, and regulates Golgi morphology, where its expression elongates the organelle ribbon structure [58,99,100,101,102]. Knockdown studies showed that GOLPH3 repression causes Golgi fragmentation and prevents cancer cell spreading. This indicates that a GOLPH3-mediated ribbon-like elongated Golgi structure may promote invasion and metastasis [103]. Additionally, Halberg et al. showed that the PITPNC1–RAB1B–GOLPH3 complex regulates Golgi elongation and enhances exocytosis of pro-invasive components, such as MMP1, FAM3C, and PDGFA, thereby driving metastatic breast, melanoma, and colon cancers [15,99]. Golgins are another family of Golgi membrane proteins that interact with Golgi-associated proteins to maintain organelle architecture [104]. Golgin-97 facilitates Golgi tethering and vesicle formation for cytokines, including IL-10 and IL-6, which alters the cancer secretome during metastasis [105].

The third morphology of the Golgi exists in a fragmented form, which is especially seen during mitosis in physiological conditions. Fragmentation of the Golgi first begins in the G2 phase of the cell cycle, where individual Golgi stacks are dissociated and then undergo fragmentation into small vesicles. These vesicles are then reformed into the Golgi in each new daughter cell during the telophase [106,107]. In advanced prostate cancer cells, the fragmented Golgi was also implicated in cancer metastasis (Figure 2). Decreased expression of the Golgi matrix proteins—Giantin, GM130, and GRASP65—as well as a disrupted interaction between RAB6a and Myosin IIA, promote Golgi fragmentation and carcinogenesis [108,109]. In addition to the previously described effects on trafficking, hypoxia and NO can also mediate Golgi fragmentation and cancer malignancy. Using a high-throughput RNAi screen, previous work implicated dual-specificity phosphatase-2 (DUSP-2) in maintaining Golgi integrity, where the loss of the protein led to Golgi disruption [110]. Lin et al. further showed that hypoxia suppresses DUSP-2 expression, leading to chemoresistance and malignancy in many cancers, including colon, lung, and breast cancer [111]. Thus, DUSP-2 might be the link that connects hypoxia and Golgi fragmentation in cancer. Conversely, Lee et al. showed that compounds that scavenge NO can also promote Golgi fragmentation and a decrease in Golgi membrane proteins, such as SNAREs and SNAPs [112,113]. The ability of the Golgi to dynamically shift between different morphologies depending on the cellular context highlights the need for future studies focused on unraveling the regulation of complex Golgi morphology and understanding how best to target Golgi structural changes in cancer progression.

## 4. Golgi-Mediated Post-Translational Modifications (PTMs)

In addition to secretion, the Golgi also modulates PTMs of proteins, which determines their activity and localization. These PTMs include glycosylation, ubiquitination, sulfation, and lipidation. Although dysregulation of the non-Golgi-regulated PTMs can promote cancer metastasis, we will be focused on the role of Golgi in regulating glycosylation in normal and malignant cells. Glycosylation is the co- or post-translational addition of a sugar to a protein or lipid molecule, thereby forming a glycoconjugate. The added sugars include N-linked glycans, O-linked glycans, glycosaminoglycans (GAGs), glycosylphosphatidylinositol (GPI) anchors, phosphorylated glycans, and mannosylation; in contrast, glycosylation of lipids leads to glycosphingolipids (GSLs) [114]. Glycosylation of proteins is a highly regulated process and is carried out by glycosyltransferases or glycosidases that localize to precise Golgi compartments. For example, early acting glycosyltransferases that help synthesize core O-glycans, such as GalNAc transferases, localize to the cis- and medial-Golgi, while late-acting proteins, such as sialyltransferases, reside in the trans-Golgi [115]. Thus, the Golgi is central for protein glycosylation processing and maturation, and disruption of the organelle can interrupt the post-translational modification process [115,116].

Aberrant glycosylation is a frequently occurring oncogenic modification that was extensively studied previously. Given the difficulty in summarizing such a vast area of exciting research, we would like to introduce the readers to other references for an extensive understanding [117,118,119]. Glycosylation patterns of malignant cells are distinct from normal cells, where aberrant modifications correlate with worse cancer patient survival [119]. These modifications alter the folding, stability, localization, and interactions of a protein and lead to disrupted cell–cell and cell–ECM adhesions, cancer cell dissemination, and metastasis. For example, one of the most common glycosylation marks in malignant cells is β6GlcNAc side-branching of N-linked structures. It was demonstrated that the inhibition of β6GlcNAc branching enhances E-cadherin-mediated cell–cell attachments and suppresses the metastatic spread of melanoma cells [120,121]. Another example is a ganglioside-specific sialidase, namely, NEU3, which was upregulated in colon cancers, where enhanced NEU3 protected malignant cells against programmed cell death [122]. Sialylation can also be regulated by other cellular factors. Cancer cells have an alkaline Golgi pH that leads to Golgi disruption and fragmentation. Moreover, this increased pH mislocalizes the glycosyltransferases from the Golgi to the endosomes, altering glycosylation patterns in malignant cells and enhancing the Tn antigen, which is an amino acid-carbohydrate glycoconjugate [123]. The Tn antigen associated with tumor cells is truncated compared to normal cells and is called sialyl Tn. This modification allows tumor cells to interact with galectins in the endothelium, thereby promoting lymphatic invasion and lymph node metastasis of breast cancer cells [124]. Another cellular condition that can also affect glycosylation is hypoxia. Hypoxia-inducible factors (HIF1-3) that are activated during hypoxia regulate Golgi-associated glycosyltransferases (MGAT2, MGAT3, fucosyltransferases, and sialyltransferases) [73]. For example, HIF-1α-mediated suppression of FUT1 and FUT2 is favored in malignant pancreatic cancer, showing a decrease in surface α1,2-fucosylated glycan structures [125]. Additionally, FUT4 also correlates with a worse prognosis in lung cancer patients and upregulates EMT, vesicular trafficking, and lung cancer invasion [126]. Furthermore, Golgi proteins can also modulate cancer cell glycosylation by altering Golgi dynamics. The knockdown of Golgi reassembly-stacking proteins, such as GRASP55 and 65, leads to a global decrease in the abundance and diversity of cell surface glycoproteins. It is hypothesized that this loss, which unlinks the Golgi ribbon structure, can either enhance vesicle budding to reduce the time for PTM processing or can disrupt the localization of the glycosyltransferases to prevent glycosylation. In addition to affecting protein PTMs, a loss of GRASP55 also abrogates GSL biosynthesis [127]. Enzymes involved in GSL biosynthesis interact with GRASP55 and are retained in the trans-Golgi where they function. Other Golgi tether proteins, such as golgins, can similarly alter glycosylation by regulating Golgi morphology. Loss of a golgin protein, namely, giantin, in zebrafish shows large-scale changes in the expression of 22 Golgi-resident glycosyltransferases but has no effect on the ER glycosylation machinery [128]. Additionally, giantin is necessary for the function of two core glycosyltransferases, namely, C2GnT-L and C2GnT-M, as well as the extracellular expression of hyaluronan, which is a GAG that helps form the ECM. The suppression of the previously described golgins, namely, giantin and GM130, and GRASP65 not only prompts Golgi fragmentation but also prevents glycosylation processing. Thus, prostate cancer cells, where these proteins are most affected, have high cell surface mannose N-glycans, which are normally lost during Golgi-mediated processing [129]. These changes in glycosylation on the surface of the cell can alter its interaction with other cells in the TME or with components of the ECM to eventually drive metastasis. GAGs, such as chondroitin sulfate (CS), heparan sulfate (HS), and keratan sulfate, are critical modifications facilitating the formation of the ECM and are known to assist metastatic outgrowth. The invasive potential of a cell depends on the number of HS chains modifying syndecan-1, which is a plasma membrane proteoglycan and a major factor in invasion and metastasis in myeloma and colorectal cancer [130,131]. CS is another GAG that correlates with worse patient survival by activating MMPs, altering the ECM, and promoting metastasis in various cancer types [132].

The specific proteoglycome of malignant cells promoted by changes in the Golgi, as well as in Golgi-associated proteins, designate the glycosylation signature of cancer cells. Hence, this signature can be used as a biomarker to identify tumorigenic cells. For example, tumor-associated Tn may be used as a biomarker for malignant cells, as it is expressed in more than 90% of breast tumors, and 70–80% of lung, colon, cervix, bladder, prostate, ovarian, and stomach cancers [124]. The Tn antigen also correlates with metastasis and significantly worse patient prognosis. Another glycosylation signature that may be used as a biomarker is the high mannose N-glycans in prostate cancer or the increased branching of N-linked glycans, such as β1,6-GlcNAc, in breast cancer [133]. In Kaposi’s sarcoma infected with Kaposi’s-sarcoma-associated herpesvirus, the viral IL-6 is differentially N-glycosylated compared to the host IL-6, which contributes to enhanced activation of JAK/STAT signaling and B-cell proliferation [134]. This study suggested that glycosylation of secreted proteins alters tumor progression and could potentially be used as biomarkers. Further studies in identifying the specific proteoglycomic signature of specific tumor types may help in the early detection and treatment of cancer.

## 5. Therapeutic Targeting of Tumor Secretion

Therapeutics targeting protein secretion is an attractive approach to inhibit the spread of cancer as inhibition of secretory pathways can prevent the deposition of ECM components or pro-tumorigenic enzymes that remodel the TME. The Golgi already has a host of drugs that target aspects of the organelle, thus limiting and inhibiting cancer-mediated secretion [135]. Small GTPase ARF1 inhibitors are the most widely researched therapeutics targeting the Golgi. Brefeldin-A (BFA), which is a fungal toxin, blocks ARF1 activity by interfering with GBF-1, which is an ARF-GEF, disrupting Golgi-mediated secretion, promoting apoptosis, and inhibiting tumorigenesis and metastasis. Although BFA has worked well to limit cancer progression in murine models in vitro and in vivo, its clinical potential has been limited. BFA has low solubility, inadequate bioavailability following oral dosage, and fast clearance from the blood [135]. The inhibitor GRASPIN lowers metastatic burden in NSCLC murine models by targeting the Golgi-associated protein G55, which aids in the secretion of pro-tumorigenic factors, such as IGFBP2 and SPP1 [136]. PIK93 targets a Golgi-associated protein, namely, PI4KIIIβ; however, the drug also recognizes class I and class III PI3Ks due to the conservation of the phosphoinositide kinase domain, which is involved in a variety of other functions. To overcome this non-specificity, drugs have been developed, such as IN-9, with strong specificity for PI4KIIIβ, where this inhibition increases murine lung tumor cell apoptosis and disrupts Golgi-mediated secretion [13,137]. Additionally, vesicular trafficking proteins can also be targeted. NAV-2729 specifically blocks ARF6 function [138]. Inhibitors against RAB proteins, such as Pitavastatin, target HMG CoA-reductase, which prevents membrane localization and activation of RAB proteins by blocking prenylation [49]. This method is commonly used due to the difficulty in developing inhibitors against the GTPase activity of specific RAB proteins, as this region is highly conserved across RAB family members. Another method to target RAB proteins includes inhibiting the interaction between RAB proteins and their effectors. One example is RFP26, which blocks RAB11 from binding to its effectors [49]. A synergistic therapeutic strategy may exist with the use of RAB inhibitors and actin or myosin inhibitors, as vesicles guided by RAB proteins commonly traffic within the cell using actin or myosin [40]. For example, the use of actin inhibitors disrupts the trafficking of a drug efflux pump in breast cancer cells and enhances the efficacy of chemotherapy using daunorubicin [139]. Secreted proteins, such as MMPs, and Golgi proteins involved in post translational modification can also be targeted for potential therapies against altered Golgi function [140,141,142,143]. These reviews further summarize drugs targeting the Golgi and secretion in cancer beyond those already discussed here [135,144].

Despite the remarkable tumor regression in preclinical models, there are complications in translating these types of drugs that target secretion into the clinic. In addition to the abovementioned difficulties, compensation can also occur as multiple secretory pathways have overlapping functions. For example, the autophagy pathway is also responsible for the secretion of certain proteins, such as MMPs and ECM proteins. This is termed secretory autophagy and is commonly activated in cancers. Moreover, another type of unconventional protein secretion, namely, type IV UPS, involves protein secretion from the ER to the PM without passage through the Golgi. In type IV UPS, proteins with a signal peptide or transmembrane domain circumvent the Golgi, as they are trafficked in the anterograde direction from the ER to the PM or ECM. The pathway is usually activated upon ER stress—such as nutrient deprivation, hypoxia, and increased metabolism—all of which are instigators of malignancy [145]. Type IV UPS proteins are secreted during tumorigenesis, even in the presence of a Golgi inhibitor, namely, BFA, which indicates that cancer cells adopt a bypass mechanism [146]. It will be necessary to study the intricate interactions between CPS and UPS to develop effective therapies that fully inhibit secretory pathways that can promote cancer metastasis. Finally, the dynamic nature of the Golgi structure also poses challenges when using it to therapeutically inhibit cancer-mediated secretion [103]. As discussed before, the multiple Golgi architectures (Figure 2) can act as drivers of cancer metastasis that are facilitated by different Golgi-associated oncoproteins [91,100]. Hence, targeting the functional morphology of the organelle is a fine-tuned balance and requires a deeper investigation into key players of the process.

## 6. Conclusions

The Golgi apparatus serves many vital, conserved functions in the cell, such as the regulation of protein trafficking, post-translational modification, and secretion. Hence, mutations in Golgi-associated proteins alter Golgi orientation and morphology, facilitating directed secretion of pro-metastatic factors. Much remains to be discovered regarding the mechanisms of Golgi-mediated exocytosis and how this process promotes cancer metastasis at the intracellular and extracellular levels. To identify cancer-specific secreted proteins, the development of methodologies, with higher sensitivity than those currently available, is required; this will also aid in studying the dynamic nature of the Golgi and interconnected secretory pathways. Cancer-mediated secretion and secretome research is gaining the interest of the scientific community to identify novel anti-cancer therapies that will be able to slow the metastatic progression of cancer and enhance patient survival.

## Figures and Tables

**Figure 1 cells-11-01484-f001:**
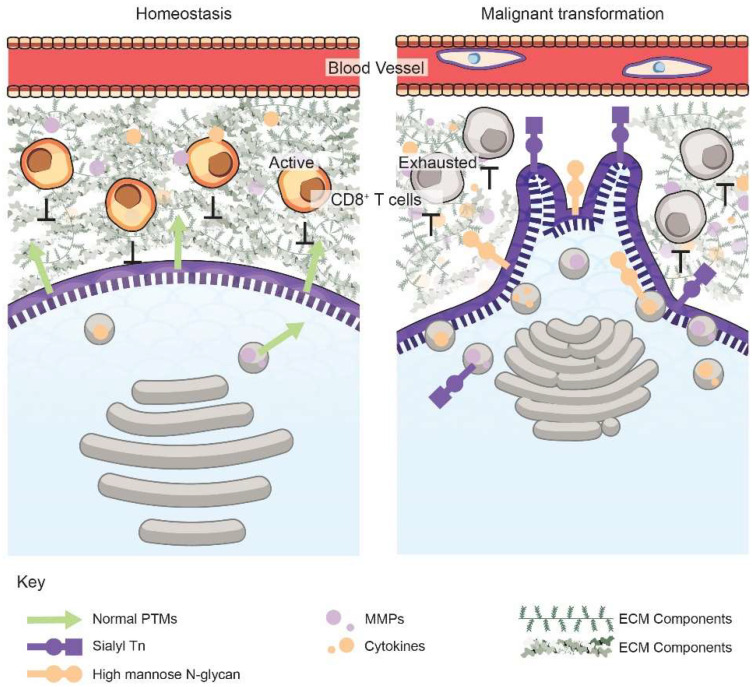
Golgi orientation, vesicular trafficking, and secretion are dysregulated upon malignant transformation. The functions, morphology, and orientation of the Golgi within a cancer cell (right) are modified, resulting in targeted exocytosis of invasive components, such as MMPs (purple dots) or cytokines (orange dots). This secretion mediates degradation of the ECM comprising of collagens, proteoglycans, etc., as well as suppression of the immune landscape. Malignant cells also have altered glycobiology and glycocalyces, such as specific surface PTMs, including sialyl Tn (purple square and circle) and high mannose N-glycans (orange circles), which promote invasion and intravasation. The unique proteoglycans and PTMs on cancer cells can serve as diagnostic or prognostic biomarkers for early cancer detection or personalized therapeutics.

**Figure 2 cells-11-01484-f002:**
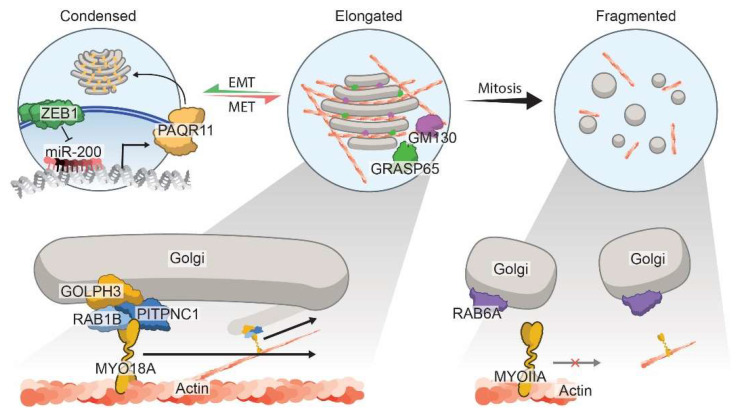
Regulation of Golgi morphology states in cancer. The Golgi oscillates dynamically between three main defined morphological states: the condensed (left Golgi), the elongated (middle Golgi), or the fragmented (right Golgi). Zeb1-mediated EMT orchestrates Golgi condensation by repressing the epithelial miRNA, namely, miR-200, and upregulating PAQR11. PAQR11 is a Golgi scaffold protein that mediates Golgi organization and secretion of pro-metastatic factors. The elongated Golgi morphology is maintained via the PITPNC1–RAB1B–GOLPH3–MYO18A complex, where GOLPH3 links vesicles to myosin motors and facilitates movement along actin filaments. Additionally, other Golgi matrix proteins, namely, GM130 and GRASP65, also interact with various Golgi factors to promote the elongated ribbon-like structures, which is necessary for cancer cell invasion and vesicular trafficking. Loss of these matrix proteins prevents organelle stacking, thereby leading to a fragmented Golgi architecture, which is commonly seen during mitosis. Moreover, in prostate cancer, a disrupted Golgi promotes invasion through dysregulated vesicle interaction with RAB6A, which results in diminished interactions with myosin, MYOIIA, and fewer fusion events.

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
