# Peer review of "Dance of The Golgi: Understanding Golgi Dynamics in Cancer Metastasis"

_cells, 2022, doi:10.3390/cells11091484_

Round 1

Reviewer 1 Report

Review of Bajaj et al

The authors provide a summary of some of the changes observed in the Golgi apparatus in cancer cells. They do this by first providing an overall context of Golgi structure and function and then expand their discussion into changes in Golgi proteins, trafficking through the Golgi and consequences of these changes in cancer cells as regards epithelial-to-mesenchymal transition and metastasis. Some of the data discussed comes from recent work from the author’s own lab. A consequence is that, conceptually, the discussion remains in the realm of what has been observed in their own papers and does not extend into scholarship with respect to how the Golgi apparatus dances in cancer cells. Thus overall, this review does not provide the reader a summary of the very powerful data that exist already with respect to this dance. This review remains at the level of an average recitation which does not do justice to the research field. There are large gaps in academic scholarship which must be filled. Some of these are:

  1. It is notable that throughout the authors use the term “Golgi” to, in reality mean, the “Golgi apparatus.” May I instruct the authors that that apparatus was called “apparato reticular de Golgi” by someone as famous as Ramon y Cajal in 1914. Out of respect for Dr. Camillo Golgi, one must use the term “apparatus” at least once in the manuscript, and then proceed to use whatever abbreviation. This reviewer cannot let pass a manuscript which simply states “Golgi” throughout. The authors may not know that on the day that Dr. Golgi died in the 1926, it was unclear whether the apparatus existed at all – it was called an artifact. Dr. Golgi, yes, he existed. The “apparatus” did not come to be accepted until 1952. Please have some respect for the long and difficult march of science in your word usage.
  2. It is also notable that lines 378-381 recite that glycosylation patterns change in and on cancer cells. This has been known since 1979. Yet the authors do not provide even one single citation to the spectacular work of Dr. S. Hakomori in uncovering these features of cancer cells, mechanisms mediated by the Golgi apparatus. There are approx. 48 papers authored by Dr. Hakomori, and many hundreds more by others elucidating the same in detail. This reviewer cannot let pass a manuscript which provides no relevant citations to Dr. Hakomori’s work over the decades. Please provide the reader an entrance into that literature.
  3. The authors show in Fig. 1 that the cisternae are spaced apart in normal cells, but are stacked closely in cancer cells. Yet there are absolutely no citations to any EM work on the Golgi apparatus in cancer cells. A detailed summary of EM studies of the Golgi apparatus as these exist inside cancer cells in actual tumors is required. I am aware that such details exist in the literature.
  4. The authors must look up and summarize the role of IL-6 in the tumor secretome and tumor EMT. Moreover, IL-6 is secreted in different glycosylation states by different cancer cells. The authors need to do a deep dive into this literature.
  5. There are additional changes in tumors that drive alterations in the Golgi apparatus in cancer that are completely ignored by the authors. Hypoxia and nitric oxide levels drive structural and functional changes in the Golgi apparatus in cancer. These must be summarized. NO scavenging markedly fragments the GA. The authors must look up the literature.
  6. BTW hypoxia and NO scavenging lead to increased endothelial cell size (“megalocytosis”) which leads to blockage of pulmonary arterial lumen and a fatal disease called pulmonary hypertension. Histologically, this disease is characterized by quasi-cancer changes including EMT – and dramatic changes in Golgi apparatus (GA). Even more specific, the compound monocrotaline pyrrole triggers these changes in GA and causes massive enlargement and dispersion of Golgi cisternae, with consequent changes in the secretome, and the endothelial and vascular smooth muscle surface. The authors must look up that literature, and summarize it. This leads to my next point.
  7. There is something big and forward-looking missing in this manuscript. There is absolutely no discussion of changes within cancer cells of Golgi tethers, SNAREs and SNAPs. These are the components of the actual machinery that make membrane vesicles move around. The field of cell biology has moved to the level of discussing disease-related changes at the levels of tethers, SNAREs and SNAP proteins. What are the changes observed in cancer cells in these molecules – which, in turn, drive the Golgi dance? For comparison please look up the literature in pulmonary hypertension/EMT and Golgi changes.

Overall, there are many large areas of science relevant to the subject matter that have been ignored by these authors.

Author Response

We would like to sincerely thank the reviewers for their careful and critical analysis of the manuscript. To further solidify the review, we have included the information suggested by the reviewers and modified the manuscript as outlined in the point-by-point rebuttal below. We believe that the inclusion of the suggested changes and additional scientific topics has greatly improved the manuscript.

Review of Bajaj et al. – Reviewer 1

The authors provide a summary of some of the changes observed in the Golgi apparatus in cancer cells. They do this by first providing an overall context of Golgi structure and function and then expand their discussion into changes in Golgi proteins, trafficking through the Golgi and consequences of these changes in cancer cells as regards epithelial-to-mesenchymal transition and metastasis. Some of the data discussed comes from recent work from the author’s own lab. A consequence is that, conceptually, the discussion remains in the realm of what has been observed in their own papers and does not extend into scholarship with respect to how the Golgi apparatus dances in cancer cells. Thus overall, this review does not provide the reader a summary of the very powerful data that exist already with respect to this dance. This review remains at the level of an average recitation which does not do justice to the research field. There are large gaps in academic scholarship which must be filled. Some of these are:

We greatly appreciate the author’s constructive comments in support of the manuscript and have addressed the detailed concerns below.

  1. It is notable that throughout the authors use the term “Golgi” to, in reality mean, the “Golgi apparatus.” May I instruct the authors that that apparatus was called “apparato reticular de Golgi” by someone as famous as Ramon y Cajal in 1914. Out of respect for Dr. Camillo Golgi, one must use the term “apparatus” at least once in the manuscript, and then proceed to use whatever abbreviation. This reviewer cannot let pass a manuscript which simply states “Golgi” throughout. The authors may not know that on the day that Dr. Golgi died in the 1926, it was unclear whether the apparatus existed at all – it was called an artifact. Dr. Golgi, yes, he existed. The “apparatus” did not come to be accepted until 1952. Please have some respect for the long and difficult march of science in your word usage.

We appreciate the author’s note that we must respect the work of the scientific leaders on the Golgi apparatus. Hence, we have introduced the Golgi as the “Golgi apparatus” in our introduction (lines 66-68) and have also cited the paper that celebrates Drs. Camillo Golgi and Ramon y Cajal for their contribution to the field.

  1. It is also notable that lines 378-381 recite that glycosylation patterns change in and on cancer cells. This has been known since 1979. Yet the authors do not provide even one single citation to the spectacular work of Dr. S. Hakomori in uncovering these features of cancer cells, mechanisms mediated by the Golgi apparatus. There are approx. 48 papers authored by Dr. Hakomori, and many hundreds more by others elucidating the same in detail. This reviewer cannot let pass a manuscript which provides no relevant citations to Dr. Hakomori’s work over the decades. Please provide the reader an entrance into that literature.

We acknowledge the reviewer’s comment and have attempted to address this point by multiple approaches. Given the amount of information regarding glycosylation in cancer cells, it is difficult to summarize it all in a review that is focused on Golgi dynamics in cancer metastasis. Thus, given the importance of the Golgi apparatus in glycosylation and cancer, we have cited the work by Dr. Hakomori in lines 461-464 for readers wanting an in-depth understanding. Moreover, we have also discussed additional work from Dr. Hakomori and his colleagues on the cancer glycome in lines 468-473.

  1. The authors show in Fig. 1 that the cisternae are spaced apart in normal cells, but are stacked closely in cancer cells. Yet there are absolutely no citations to any EM work on the Golgi apparatus in cancer cells. A detailed summary of EM studies of the Golgi apparatus as these exist inside cancer cells in actual tumors is required. I am aware that such details exist in the literature.

We thank the reviewer for this important point. We have addressed it by discussing the methods used to study the Golgi and its functions in lines 95-107. In addition to electron microscopy, we have also detailed other assays that are used to study the Golgi. We have cited these references for the readers, which include detailed structural analysis of how the Golgi changes within tumor tissues and cells.

  1. The authors must look up and summarize the role of IL-6 in the tumor secretome and tumor EMT. Moreover, IL-6 is secreted in different glycosylation states by different cancer cells. The authors need to do a deep dive into this literature.

We value this comment regarding the multiple impacts of IL-6 on the tumor microenvironment (TME) and EMT. We have discussed the role of various cytokines in altering the TME and the cancer cell secretome in this review. Given that there are many critical cytokines involved in driving cancer metastasis, including IL-6, we have only briefly addressed and referenced the key interleukins regarding their general functions in the TME. To address the reviewer’s specific comment regarding the importance of IL-6 in cancer, we have further described the ubiquitous role of the interleukin in various cancer types and patient survival, as well as in EMT-mediated invasion (lines 168-171). The reviews cited go into a more in-depth analysis of the proteins and can be referenced by the readers. Furthermore, we have also addressed the reviewer’s comment on discussing the effect of different glycosylation states of IL-6. Lines 539-543 deliberate on how different glycosylation states of IL-6 are associated with signaling pathways that can promote cancer and could be used for future biomarker studies.

  1. There are additional changes in tumors that drive alterations in the Golgi apparatus in cancer that are completely ignored by the authors. Hypoxia and nitric oxide levels drive structural and functional changes in the Golgi apparatus in cancer. These must be summarized. NO scavenging markedly fragments the GA. The authors must look up the literature.
  2. BTW hypoxia and NO scavenging lead to increased endothelial cell size (“megalocytosis”) which leads to blockage of pulmonary arterial lumen and a fatal disease called pulmonary hypertension. Histologically, this disease is characterized by quasi-cancer changes including EMT – and dramatic changes in Golgi apparatus (GA). Even more specific, the compound monocrotaline pyrrole triggers these changes in GA and causes massive enlargement and dispersion of Golgi cisternae, with consequent changes in the secretome, and the endothelial and vascular smooth muscle surface. The authors must look up that literature, and summarize it. This leads to my next point.

For point 5, we appreciate the reviewer for pointing out this oversight on our part. Hypoxia and nitric oxide are indeed critical drivers of changes in the Golgi apparatus and cancer metastasis. Hence, to enhance the impact of our review, we have presented an in-depth discussion of how both these cellular conditions, hypoxia and NO scavenging, can alter trafficking (lines 292-307), the Golgi apparatus morphology (lines 417-426), and glycosylation (Lines 481-489).

We thank the reviewer for this comment in point 6 and for expanding our knowledge on how hypoxia and NO affect the Golgi in quasi-cancer related diseases, such as pulmonary hypertension. While this is an intriguing point-of-view, we believe that it is prudent to emphasize and keep the focus of the review on how hypoxia and NO scavenging can alter the Golgi apparatus and the metastatic secretome.  Given the focus on cancer metastasis in this review, we kindly think that other diseases such as pulmonary hypertension may be beyond the scope of this paper; thus, we have not discussed the role of hypoxia and NO in pulmonary hypertension.

  1. There is something big and forward-looking missing in this manuscript. There is absolutely no discussion of changes within cancer cells of Golgi tethers, SNAREs and SNAPs. These are the components of the actual machinery that make membrane vesicles move around. The field of cell biology has moved to the level of discussing disease-related changes at the levels of tethers, SNAREs and SNAP proteins. What are the changes observed in cancer cells in these molecules – which, in turn, drive the Golgi dance? For comparison please look up the literature in pulmonary hypertension/EMT and Golgi changes.

The reviewer’s comment on including the SNARE and SNAP proteins in our discussion is much appreciated as it will improve our understanding of the Golgi dance. To address this, we have delved deep into how the proteins are altered in various cancer types and how this affects Golgi trafficking and promotion of cancer metastasis (lines 247-267). For reasons described above, we have not elaborated on their role in pulmonary hypertension.

Reviewer 2 Report

I think this is an interesting review that deals with the role of the Golgi complex and membrane traffic in some specific pathological conditions affecting cancer. The manuscript is well written and well balanced and offers even non-expert readers the opportunity to understand the topics covered. Therefore, I think it will be of interest to the readers of cells.
However, before publication, I have some suggestions on some aspects that are unclear to me.
The first suggestion is related to the figure describing the alterations of the Golgi structure in some pathological conditions (Figure 2). An aspect that can be confusing is the term mitosis in the figure. The text mentions structural alterations resulting from the altered expression of GM130 or GRASP65, not mitosi.

In addition, the authors report important examples in which the alteration of a set of proteins  (e.g.,  ZEB1) induces a modification of the structure and an important functional consequence (eg. Golgi compaction and enhanced exocitosis). My perplexity derives from the fact that the structural alteration of the Golgi and the functional consequence may not necessarily be linked by a cause and effect relationship, but they could be independent events. In other words, can the statement "These morphologies have different described effects on cancer metastasis" be generalized? (lines 296-297).

As a final suggestion, I recommend adding some references and checking others.

Line 121. Check reference 21

Lines 252-252: it is not clear to me if the statement refers to reference 52 or if another one should be added

Line 267, I could be wrong, but it seems to me that the only recognized oncogene associated with Golgi is GOLPH3

Evaluate whether to add references to the statements described in lines: 171-172; 275-277, 283-284

Finally, in the paragraph related to GOLPH3 I think it is important also to mention:
Rizzo et al., EMBO J; 2021 Apr 15; 40 (8): e107238. doi: 10.15252 / embj.2020107238.

Author Response

Review of Bajaj et al. – Reviewer 2

I think this is an interesting review that deals with the role of the Golgi complex and membrane traffic in some specific pathological conditions affecting cancer. The manuscript is well written and well balanced and offers even non-expert readers the opportunity to understand the topics covered. Therefore, I think it will be of interest to the readers of cells. However, before publication, I have some suggestions on some aspects that are unclear to me.

We appreciate the author’s comments supporting the manuscript and believe that the changes, as detailed below, appropriately clarify the major and minor points.

  1. The first suggestion is related to the figure describing the alterations of the Golgi structure in some pathological conditions (Figure 2). An aspect that can be confusing is the term mitosis in the figure. The text mentions structural alterations resulting from the altered expression of GM130 or GRASP65, not mitosis.

We acknowledge the concerns raised by the reviewer and would like to clarify a few details to address this point. Mitosis is a physiological condition where the Golgi is completely fragmented, as the cells divide to form daughter cells. Hence, a similar process could be involved in tumorigenesis where the Golgi apparatus is disintegrated, as shown in Figure 2. To clarify our intent for the reader, we have further elaborated on this in lines 409-413.

  1. In addition, the authors report important examples in which the alteration of a set of proteins (e.g., ZEB1) induces a modification of the structure and an important functional consequence (e.g. Golgi compaction and enhanced exocytosis). My perplexity derives from the fact that the structural alteration of the Golgi and the functional consequence may not necessarily be linked by a cause and effect relationship, but they could be independent events. In other words, can the statement "These morphologies have different described effects on cancer metastasis" be generalized? (lines 296-297).

We greatly appreciate the author’s note that the functional consequence may not be a “cause-and-effect” phenomenon. Hence, we have addressed this issue by changing our wording to one that shows a correlative relationship. This is apparent in lines 363, and 380-382.

  1. Line 121. Check reference 21
  2. Lines 252-252: it is not clear to me if the statement refers to reference 52 or if another one should be added
  3. Line 267, I could be wrong, but it seems to me that the only recognized oncogene associated with Golgi is GOLPH3
  4. Evaluate whether to add references to the statements described in lines: 171-172; 275-277, 283-284
  5. Finally, in the paragraph related to GOLPH3 I think it is important also to mention: Rizzo et al., EMBO J; 2021 Apr 15; 40 (8): e107238. doi: 10.15252 / embj.2020107238

We appreciate the reviewer pointing out these oversights on our part, and we have addressed all these points in our revisions.

Point 3 – Lines 140-141. Reference 30.

Point 4 – Lines 315-316. Reference 72.

Point 5 – Line 330. We only talk about Golgi proteins, not Golgi oncoproteins. We have changed the terminology we used for the sake of clarity.

Point 6 – Lines 192-194/ Reference 40. Lines 338-339/ Reference 86. Lines 349-351/ Reference 90.

Point 7 – This was a very interesting paper that highlighted how in addition to all its other functions, GOLPH3 can also regulate trafficking of glycosphingolipids to the plasma membrane, thereby mediating signaling of mitogenic pathways. We appreciate the reviewer bringing this paper to our attention. We believe that the addition of this information further emphasizes the multi-faceted and critical role of GOLPH3 and the Golgi apparatus changes in carcinogenesis (lines 280-284).